# Nuclear Receptors Linking Metabolism, Inflammation, and Fibrosis in Nonalcoholic Fatty Liver Disease

**DOI:** 10.3390/ijms23052668

**Published:** 2022-02-28

**Authors:** Tobias Puengel, Hanyang Liu, Adrien Guillot, Felix Heymann, Frank Tacke, Moritz Peiseler

**Affiliations:** 1Department of Hepatology & Gastroenterology, Charité Universitätsmedizin Berlin, Campus Virchow-Klinikum and Campus Charité Mitte, 13353 Berlin, Germany; tobias.puengel@charite.de (T.P.); hanyang.liu@charite.de (H.L.); adrien.guillot@charite.de (A.G.); felix.heymann@charite.de (F.H.); 2Berlin Institute of Health (BIH), 10178 Berlin, Germany

**Keywords:** nonalcoholic steatohepatitis, nuclear receptors, NAFLD, inflammation, macrophages, PPAR, FXR

## Abstract

Nonalcoholic fatty liver disease (NAFLD) and its progressive form nonalcoholic steatohepatitis (NASH) comprise a spectrum of chronic liver diseases in the global population that can lead to end-stage liver disease and hepatocellular carcinoma (HCC). NAFLD is closely linked to the metabolic syndrome, and comorbidities such as type 2 diabetes, obesity and insulin resistance aggravate liver disease, while NAFLD promotes cardiovascular risk in affected patients. The pathomechanisms of NAFLD are multifaceted, combining hepatic factors including lipotoxicity, mechanisms of cell death and liver inflammation with extrahepatic factors including metabolic disturbance and dysbiosis. Nuclear receptors (NRs) are a family of ligand-controlled transcription factors that regulate glucose, fat and cholesterol homeostasis and modulate innate immune cell functions, including liver macrophages. In parallel with metabolic derangement in NAFLD, altered NR signaling is frequently observed and might be involved in the pathogenesis. Therapeutically, clinical data indicate that single drug targets thus far have been insufficient for reaching patient-relevant endpoints. Therefore, combinatorial treatment strategies with multiple drug targets or drugs with multiple mechanisms of actions could possibly bring advantages, by providing a more holistic therapeutic approach. In this context, peroxisome proliferator-activated receptors (PPARs) and other NRs are of great interest as they are involved in wide-ranging and multi-organ activities associated with NASH progression or regression. In this review, we summarize recent advances in understanding the pathogenesis of NAFLD, focusing on mechanisms of cell death, immunometabolism and the role of NRs. We outline novel therapeutic strategies and discuss remaining challenges.

## 1. Introduction

The prevalence of nonalcoholic fatty liver disease (NAFLD) has increased profoundly over the last decades-to date, representing the most common cause of chronic liver disease worldwide and numbers are expected to increase even further [1,2]. NAFLD is closely connected to the metabolic syndrome including obesity, hypertension, insulin resistance and dyslipidemia. NAFLD is characterized by lipid and lipid metabolite accumulation in hepatocytes, and a general disruption of liver homeostasis and metabolism. Of note, NAFLD is not only a consequence of fat accumulation but might also be a major driver for the development of the metabolic syndrome and its complications [3,4]. During NAFLD, chronic low-grade sterile inflammation-often termed metainflammation-leads to the development of a nonalcoholic steatohepatitis (NASH), which can progress to liver fibrosis, cirrhosis, and carries the risk of associated hepatocellular carcinoma (HCC) [5,6]. Of note, HCC in the context of NAFLD can occur even in the absence of cirrhosis [7]. Liver fibrosis is the key determinant of liver-related and overall mortality in patients, projecting a tremendous burden to people’s health and our health care systems as total numbers, as well as proportion of patients with liver fibrosis, are constantly rising [8,9,10]. In contrast to the (decompensated) cirrhotic disease stage, NASH and liver fibrosis are considered reversible and thus potentially treatable. Although basic and translational research have identified many potential drug targets, approved pharmacological compounds for the treatment of NASH and fibrosis are still lacking, reducing current treatment strategies to difficult-to-sustain weight loss and lifestyle changes or to managing comorbidities such as optimizing diabetes therapy [11].

The pathogenesis of NAFLD is complex and still incompletely understood, owing to the multisystem character of the disease. Numerous disease-promoting mechanisms have been identified, such as genetic risk profile, environmental factors, metabolic dysregulation, inflammation, intestinal dysbiosis, molecular factors (oxidative stress, lipotoxicity, mechanisms of cell death) and fibrogenesis (Figure 1) [12]. Due to the multiplicity and heterogeneity of these factors, a personalized patient-based approach is probably needed for effective NAFLD/NASH therapy. Accordingly, an ideal treatment for NAFLD would target (early) lipid overload and the ensuing cell stress and cell death, and subsequently dampen inflammation and fibrosis (Figure 1) [13]. Novel strategies that have been proposed include either combination therapy of agents targeting multiple components, such as inflammation and metabolism, or single drugs with multiple targets [13]. With a diverse array of potential mechanisms of action, nuclear receptors (NRs) provide an attractive drug target in NAFLD, potentially modulating deranged metabolism and key inflammatory pathways. NRs are a family of transcription factors, activated by a variety of ligands including hormones, lipids and bile acids [14]. Peroxisome proliferator-activated receptors (PPARs), liver X receptors (LXRs) and farnesoid X receptor (FXR) are among the most important NRs with respect to NAFLD (as discussed in this review). 

However, the importance of NR by far extends its immediate functions in hepatocytes, as they also affect responses in non-parenchymal liver cell populations. In the last decades, macrophages emerged as key players in chronic liver diseases such as NAFLD. Translational and clinical studies supplanted the former paradigms of “pro- and anti-inflammatory M1/M2 macrophages”, by revealing a tremendous heterogeneity among macrophage subsets and highlighting their functional plasticity [15]. Macrophages can adapt to a wide range of stimuli including metabolic factors leading to a metabolic cellular reprogramming, modulating disease progression at multiple stages and in various tissues [16,17,18,19]. For instance, macrophages were identified as key drivers in the regulation of inflammation-associated insulin signaling [20,21]. Therefore, further investigations analyzing the complex interactions between disease-dependent metabolic dysregulation and immune cells are urgently needed. Closing the loop, NR ligands, such as PPAR agonists, were shown to dampen inflammatory macrophages in NAFLD in addition to restoring metabolic homeostasis, thus offering a therapeutic link between correction of metabolic dysfunction and alleviation of inflammatory insults [22]. In addition, the activation of hepatic stellate cells (HSCs) as the main collagen-producing myofibroblast population promoting liver fibrosis, is also influenced by NRs [23]. In the present review, we recapitulate some established concepts and new insights into the pathogenesis of NAFLD. We highlight key triggers of liver inflammation in NASH, including different mechanisms of cell death and their impact on the activation phenotype and metabolic reprogramming of key immune cells (i.e., liver macrophages). We explore the roles of NRs in NAFLD and conclude with a discussion on promising therapeutic strategies aimed at targeting metabolic pathways in NAFLD. 

## 2. Cellular Stress, Hepatic Cell Death and Inflammatory Modulation in NAFLD

During progression towards more advanced stages of NAFLD, including NASH, fibrosis, cirrhosis and NASH-associated HCC, inflammation plays a pivotal role [24]. Hepatocyte stress and cell death have been identified as key events triggering and sustaining liver inflammation in NASH. Lipid deposition in hepatocytes leads to lipotoxicity, mitochondrial dysfunction, production of reactive oxygen species (ROS), and endoplasmic reticulum (ER) stress [25]. During NASH, the inflammatory environment in the liver is composed of multiple innate and adaptive immune cells. A vicious cycle ensues, as inflammatory cytokines produced by immune cells increase metabolic dysregulation in adipose tissue and in the liver, resulting in additional stress-induced cell death [26]. 

### 2.1. Cellular Stress

Lipid-mediated cellular stress, potentiated by cytokines released by infiltrating myeloid cells, gradually causes liver damage. Indeed, steatosis induced by long-term feeding of high-fat diet (HFD) under experimental conditions in mice, increases hepatocyte susceptibility to cytokine-induced cell death [27]. Furthermore, oxidative stress is significantly involved in the NASH pathogenesis [28,29] (Figure 1 and Figure 2). Along those lines, anti-ROS treatment was reported to attenuate NASH progression in HepΔIkbkg mice [30]. Furthermore, oxidative stress during NASH significantly induced the exhaustion of tyrosine phosphatase, including tyrosine phosphatase non-receptor type 2 (PTPN2, also known as TCPTP) [31]. Deficiency of PTPN2 in hepatocytes resulted in maladjustment of signal transduction and activation of transcription 1 (STAT1) and STAT3, which eventually accelerated the progression on the spectrum of steatosis-NASH-cirrhosis-hepatocellular carcinoma (HCC) in mice. Interestingly, in this study the authors found an uncoupling of steatosis and carcinogenesis; while STAT1 activation propagated liver steatosis, STAT3 mediated HCC development. Of note, mitochondrial oxidative stress caused by the accumulation of free cholesterol enhanced the hepatic sensitivity to cytokine-induced cell death [32]. Recently, oxidized phospholipids were shown to be upregulated and enriched in human and murine NASH, resulting in mitochondrial dysfunction and further activation of reactive oxygen species (ROS) [33].

### 2.2. Hepatocyte Cell Death

Cell death is a hallmark of chronic liver injury, and directly linked to the onset of inflammation and fibrosis, which characterizes NAFLD progression. Furthermore, chronic cell death and excessive liver repair favor the development of malignant cells resulting in NASH-associated HCC. Different mechanisms of cell death have been reported in NAFLD, including apoptosis, necrosis, necroptosis, pyroptosis and ferroptosis, each following distinct patterns, while sharing many similarities and likely coexisting over the course of a chronic liver disease. 

According to evidence from mouse models, even exclusive hepatic cell death can trigger the development of NASH [34]. Clinical studies have established that hepatocellular apoptosis in NASH patients is more severe than in patients with exclusive steatosis, and correlated with increased fibrosis and inflammation [35]. Apart from apoptosis, necrosis is a driver of NASH initiation and progression [36]. Notably, during NASH progression, the suppression of TNF-induced signaling was recently shown to reduce cell death and liver damage, thereby reducing liver steatosis and fibrosis, highlighting a crucial role of inflammatory modulation in hepatic cell death mechanisms [37]. Therefore, the crosstalk between programmed cell death and inflammatory modulation characterized the pathogenesis of NASH, albeit many of the underlying molecular mechanisms remain elusive. In recent years, a better understanding of cell death mechanisms has expanded the number of known processes involved in metabolic liver diseases, from apoptosis to mitochondrial permeability transition (MPT)-driven necrosis, and necroptosis, autophagic apoptosis, and pyroptosis [38]. In NAFLD/NASH, multiple types of cell death occur simultaneously and are featured in both parenchymal and non-parenchymal cells. Subsequently, the crosstalk between immune cells and dying parenchymal cells can eventually lead to the development of NASH. 

With the progress of cell death research in the last decade, the programmed hepatic cell death has been regarded as a trigger of chronic liver inflammation and progression. Major histopathological and pathophysiological features of NASH include immune cell infiltration, intracellular lipid accumulation and cell swelling, all of which are precursors of programmed hepatocyte death [39]. Moreover, linking cell death to inflammation, NASH patients and experimental NASH animal models revealed a striking inflammasome activation in different parenchymal and non-parenchymal cells with expression of inflammasome-associated genes including nod-like receptor protein 3 (NLRP3), pro-iL-18, pro-IL-1β, ASC and caspase-1 [40]. Pyroptosis and necroptosis, as the most representative types of inflammation-modulated hepatocyte death, can recruit and stimulate immune cells by releasing a large amount of damage associated molecular patterns (DAMPs).

MPT-driven necrosis, as a type of regulated cell death, is caused by toxins- or oxidative stress-induced cell microenvironment disturbances, which lead to the sudden disability of mitochondrial membrane potential and subsequent cell membrane rupture [41]. Unlike nuclear coagulation and cell shrinkage as seen in apoptosis, necrosis is characterized by swelling and deficiency of plasma membrane integrity [42,43]. Necroptotic hepatocytes secrete multiple cytotoxic and pro-inflammatory factors and DAMPs. The MPT resulting from permeability transition pore is regarded to rely on a dimer of the ATP synthase complex, which can be switched on through interaction with the mitochondrial protein cyclophilin D (CypD). In murine models of NASH, treatment with the CypD inhibitor cyclosporin A (CsA), or CypD gene knockout (KO) prevented MPT-driven necrosis, resulting in milder steatohepatitis [44,45,46]. However, the exact mechanism and signaling pathways of MPT-driven necrosis in NASH are still unclear.

Necroptosis is a form of programmed necrosis, or inflammatory cell death, which is controlled by receptor-interacting serine/threonine-protein kinase 3 (RIPK3) and mixed lineage kinase-like (MLKL) [47]. Generally, TNF-α, oxidative stress (OS) and mammalian target of rapamycin (mTOR)/RAC serine/threonine-protein kinase (Akt) signaling pathway, regarded as necroptotic stimuli can activate three key kinases in the necroptosis pathway through a phosphorylation sequence: RIPK1, RIPK3 and MLKL. Phosphorylation of MLKL leads to the oligomerization and binding to the cellular plasma membrane, thereby destroying the membrane and stimulating the release of cellular components, including DAMPs (such as mitochondrial DNA, S100A9, HMGB-1, ATP, IL-33, etc). DAMPs trigger and exacerbate the inflammatory modulation via binding to the surface receptors of innate immune cells [48,49]. Studies have shown that necroptosis was significantly increased in the HFD-fed mouse model [50,51]. Furthermore, extensive necroptotic changes were revealed in human NASH livers [36,52]. Recent discoveries from murine models indicate that RIPK3-modulated necroptosis not only participates in alcohol-induced liver damage, but it can also cause NASH [53,54]. RIPK3 is involved in the severity of NAFLD in humans and mice and plays a key role in the management of metabolism, injury response, inflammation, fibrosis, and carcinogenesis [55]. Thus, RIPK3-targeted therapies could be a novel and promising method to treat NASH and repress disease progression [53,54]. The inhibition of RIPK1 was recently implicated to ameliorate NASH pathologic characteristics in HFD-fed mice and to reverse steatosis development through the MLKL-mediated pathway [51].

Similar to MPT-driven necrosis, DAMPs play an essential role in necroptosis-associated immunomodulation. Interleukin (IL)-1 family cytokines are deemed as crucial components of DAMPs, participating in necroptosis regulation [56]. The kinase activity of RIP1 in NASH-associated cell death was recently investigated [57]. RIP1 deficiency alleviated NASH in mice fed the HFD. Interestingly, bone marrow chimeras revealed that RIP1 activity was particularly important in bone marrow-derived macrophages, promoting the pathogenesis of experimental NASH by mediating inflammasome activation and cell death stimulation in macrophages. Furthermore, RIP1 kinase was activated in human NASH and particularly in macrophages, thus potentially serving as a therapeutic target in NASH [57]. Multiple studies revealed that necroptosis was the main source of inflammatory modulation in disease and the genetic or pharmacologic inhibition of necroptosis could be a promising therapeutic modulation [58].

Pyroptosis has been described as a lytic inflammation-regulated cell death and was initially found to occur in macrophages infected with bacteria [59,60]. Morphologically, the pyroptotic features include ion influx, cell swelling, membrane pore formation, mitochondrial depolarization, chromatin condensation and membrane rupture, resulting in cytoplasmic secretion of interleukin (IL)-1β, IL-18 and other contents, thereby triggering inflammation [61]. The cellular hallmark of pyroptosis is known as a hole formed by the amino (N-) terminal fragment of the gasdermin (GSDM) protein family, which consists of GSDM-A, -B, -C, -D, -E and DFNB59. GSDMs are activated by pro-inflammatory effectors, such as caspase-1, caspase-4/-5 and caspase-11, which exert their effector functions by cleaving GSDM proteins [62]. 

Several studies have shown that pyroptosis may play an important role in the development of NASH [63]. A recent study reported an increase of activated caspase-1 in NASH patients, both in liver and serum, which correlated with NASH severity and fibrosis stage [63,64]. Mechanistically, hepatocyte pyroptosis was induced by inflammasome activation and interestingly, this was mediated by stellate cells engulfing extracellular NLRP3 inflammasome particles [64]. Similarly, Caspase-1-dependent hepatocellular pyroptosis induced by hepatocyte-secreted NLRP3 inflammasome led to the activation of hepatic stellate cells (HSCs) and spontaneous fibrogenesis in mice [65]. The relevance of inflammasomes in the development of NASH has been emphasized in several reports [40,66]. In the methionine- and choline-deficient (MCD) diet model, livers of Nlrp3- or caspase-1-deficient mice had reduced immune cell infiltration and pro-inflammatory gene expression, and less fibrogenesis as compared to wild-type counterparts [67,68,69]. 

The immune response mediated by pyroptosis includes several pathways: the classic pathway, the non-classical pathway and the caspase-associated pathway. Furthermore, the classic pathway can be divided into the priming pathway and assemble pathway. Via these above pathways, pyroptosis can promote the accumulation of immune cells and the activation of adaptive immune responses [70]. The vigorous secretion of cytoplasmic contents (IL-1 and IL-18) from pyroptotic cells drives local and systemic inflammation. The transcription of IL-1β promotes leukocyte recruitment and T cell activation, thereby enhancing inflammation [71]. In addition, IL-18 mediates the production of interferon (IFN)- γ and of the dynamic equilibrium of T helper (Th) -1 cells, NK cells, cytotoxic T cells, and Th2 cells. Hence, IL-18 tends to prevent tissue damage from overreactive immune responses [72,73]. Interestingly, free cholesterol is elevated in NASH livers in both humans and mice and has been proposed to be a lipotoxic DAMP, inducing NLRP3 activation and pyroptosis in Kupffer cells [40,74,75].

## 3. Crossroads between Metabolism and Inflammation-Metabolic Reprogramming of Liver Macrophages

The progression of steatosis to NASH is hallmarked by liver inflammation, including infiltration of numerous different immune cell populations, such as myeloid cells, B cells, T cells and NKT cells [76]. A plethora of studies in recent years have uncovered the critical role of macrophages in metabolic liver disease. One the one hand, macrophages drive NASH as pro-inflammatory cells, leading to cell death and hepatocyte injury; on the other hand, the hypercaloric state in obesity and metabolic syndrome alters macrophage phenotypes by interfering with their metabolism and thus reprograms their functionality, further contributing to steatohepatitis [77].

### 3.1. Immunometabolism-Metabolism-Associated Changes of Macrophages

Macrophages have traditionally been grouped into two typical subtypes: pro-inflammatory (classical M1) and anti-inflammatory (alternative M2), although this classification now appears to be too simplified to fully capture macrophage diversity [78,79,80]. In principle, pro-inflammatory macrophages are stimulated by lipopolysaccharide (LPS) and interferon-γ (IFN-γ), and are important sources of secretory cytokines, such as TNFα and IL-1β. In contrast, anti-inflammatory macrophages mainly respond to and produce IL-4 and IL-13, thereby promoting the production of cytokines involved in tissue remodeling (e.g., TGF-β). The accumulation of pro-inflammatory macrophages in lipid-enriched liver has been identified as a cause of obesity-induced chronic inflammation [81]. Pro-inflammatory macrophage-derived TNF-α induces adipocytes to produce inflammatory cytokines, and promotes liver cell stress and eventually leads to cell death. Free fatty acids (FFAs) can activate the toll-like receptor (TLR) 4 on macrophages, thereby activating the NF-κB pathway and inducing pro-inflammatory cytokine release [82]. In addition, the long-chain saturated fatty acid (LSFA)-palmitic acid is considered a specific TLR4 agonist. TLR4 can regulate SFA-induced inflammation by altering the lipid metabolism of macrophages. It was shown that macrophage populations exhibit phenotypic heterogeneity and high plasticity based on their environment and intracellular signal transduction [83,84]. Enhancement of the aerobic glycolysis mediated by the mTOR/HIF-1 signaling pathway, polarized macrophages towards a pro-inflammatory phenotype [85,86]. In contrast, the anti-inflammatory phenotype was induced in adipose tissue macrophages by oxidative phosphorylation (OXPHOS) and fatty acid oxidation [87,88,89]. Therefore, mitochondrial metabolism can regulate the function of these immune cells. Of note, the transition between aerobic glycolysis and OXPHOS plays a fundamental role in macrophage polarization. In most cases, lipid metabolism of pro-inflammatory macrophages favors fatty acid (FA) synthesis and the production of intermediate derivatives that upregulate the expression of pro-inflammatory mediators. In the context of diabetes, endogenous FA synthesis can modulate the functional phenotype of macrophages [90]. In contrast, anti-inflammatory macrophages obtain energy supply through FA degradation [91]. Indeed, compared with naïve macrophages and pro-inflammatory macrophages, IL-4 induced anti-inflammatory macrophages to exert higher activities of lipolysis, fatty acid oxidation (FAO) and oxidative phosphorylation (OXPHOS) [92]. The absence and dysfunction of this metabolic pathway results in the repression of anti-inflammatory characteristics, such as the attenuation of surface marker expression (e.g., CD206, CD301, PD-L2 and RELMα) [74]. However, this remains debatable as some studies propose the pro-inflammatory polarization to be mediated by FAO [93,94]. Taken together, these fundamental principles about immunometabolism of macrophages suggest that in conditions of severely altered metabolism, such as obesity and NAFLD, dysregulated immunometabolism contributes to macrophage mediated inflammation by skewing their phenotype.

### 3.2. Metabolic Modulation of Liver Macrophages in NAFLD/NASH

Liver-resident macrophages mainly exist in form of Kupffer cells (KCs), which can be identified by expression of C-Type Lectin Domain Family 4 Member F (CLEC4F) in mice [95]. KCs are of embryonic origin and maintained locally by self-renewal, as opposed to monocyte-derived macrophages (MoMFs) originating from the bone marrow [95]. KCs are critical to maintain homeostasis of the healthy liver, act as sentinels and are activated during onset of steatohepatitis. Contrastingly, MoMFs massively accumulate in NASH and were shown to exacerbate inflammation and fibrosis. More recent evidence suggests that Kupffer cells are lost during experimental steatohepatitis and replenished by monocyte derived Kupffer cells with similar, albeit more pro-inflammatory gene expression profiles [96,97]. Furthermore, a recent study identified a subset of metabolically active Kupffer cells, termed KC2, in mice fed a high fat diet. KC2 were numerically expanded, and specific depletion of this subset reversed obesity and steatosis in mice [80].

Liver macrophages can be directly activated by adipose tissue derived FFAs through the TLR signaling pathway. For example, palmitic acid and TLR2 may synergistically induce activation of inflammasomes in KCs/macrophages and polarize KCs/macrophages towards pro-inflammatory phenotypes [98]. In addition, trans fatty acids and peroxide lipids were shown to activate KCs [99]. Adipokines, such as adiponectin and leptin also play a role in regulating inflammation in NAFLD. In the Leptin-deficient ob/ob mouse model, the lack of leptin induced pro-inflammatory phenotype and pro-fibrotic cytokine secretion in KCs [100,101]. 

Another lipotoxic molecule in NASH is free cholesterol [102,103]. Containing abundant crystallized cholesterol, MoMFs as well as KCs are surrounding dying lipid-enriched hepatocytes, forming a crown-like structure [104,105,106,107]. KCs can engulf lipoproteins through receptors such as scavenger receptor CD36, inducing a pro-inflammatory phenotype, eventually forming pro-inflammatory foam-like macrophages [108]. Of note, cholesterol-enriched KCs exhibit more response to inflammatory stimuli (e.g., LPS) [109]. These results indicate that oxidized low-density lipoproteins can alter macrophage phenotype. However, the mechanisms involved remain unclear [110]. 

During steatohepatitis, lipotoxicity combined with inflammation leads to cell damage and necrosis, which continuously causes pro-inflammatory polarization of the liver macrophage pool. Therefore, this process suggests a potential feed-forward cycle in the pathogenesis of NASH. The production of death ligands and TNFα is increased in KCs phagocytosing apoptotic bodies derived from hepatocytes [111]. Dying hepatocytes release DAMPs, which activate pattern recognition receptors (PRRs) on KCs and induce the recruitment of immune cells (such as monocytes and neutrophils), thereby enhancing inflammation [112]. Liver macrophages can also be activated by gut-derived microbiota and their products (endotoxin, etc.). In NASH, leaky gut leads to KCs and macrophages exposure to bacteria and endotoxins [113]. Bacterial lipopolysaccharide (LPS) can activate the pro-inflammatory phenotype of KCs/macrophages, thereby exacerbating the metabolic imbalance and accelerating the progression of steatosis and fibrosis [114,115,116,117].

In summary, lipotoxicity and metabolic disturbance have a profound effect on liver macrophages, altering their phenotype during chronic steatohepatitis. Research of the more recent years has uncovered a remarkable plasticity and heterogeneity of liver macrophages, increasing the number of known subsets and their functionality [76]. Many of the older studies investigating Kupffer cells in the steatohepatitis context did so without knowledge of more specific markers such as CLEC4F or the possibility of single-cell technologies. In future studies, we will learn a lot more on specific liver macrophages in NAFLD, likely disentangling some inconsistent findings with regards to phenotype switch in different models and patient cohorts. 

## 4. Nuclear Receptor (NRs) in NAFLD/NASH: Linking Metabolism and Inflammation

The pathomechanisms of NAFLD feature a number of steps–occurring simultaneously or subsequently–that together promote advanced NAFLD. As discussed, molecular mechanisms including different forms of cell death, triggered by lipotoxicity and cell stress, as well as inflammation are critical pieces. Nuclear receptors (NRs) have an abundant array of functions centering around metabolism and inflammation. In recent years, the role of NRs in liver steatosis and NASH has been investigated in many studies [14,118,119,120,121]. NRs are transcription factors regulating the expression of multiple genes by directly regulating transcriptional activities and epigenetic changes (Figure 2). Nuclear receptors act by binding to ligands to form heterodimers with retinoid X receptor (RXR) α, β, and γ (NR2B1-3) (Figure 2). The ligands of the NR subfamily include nuclear-permeable, lipophilic, endogenous substances derived from multiple nutrients [such as Fas, eicosanoids, oxysterols, bile acids (Bas)] and other exogenous chemical substances. In humans, 48 NRs were identified thus far [122]. It is notable that NR functions are related to energy/nutrient management, which may play a critical role in the pathogenesis of NAFLD [14]. However, NRs are also expressed by critical immune cells, including macrophages, resulting in direct regulation of inflammation. Hence, targeting NRs might positively influence the course of chronic steatohepatitis by modulating one or more of the disease-promoting avenues (Figure 1), i.e., lowering metabolic stress in hepatocytes, while simultaneously promoting anti-inflammatory macrophages. In addition, shielding hepatocytes from metabolic stress can also indirectly dampen pro-inflammatory immune cells, providing an additional stimulus for anti-inflammatory cells. Here, we will introduce the role of several NRs in the pathophysiology of NAFLD/NASH and discuss their preventive and therapeutical potential. 

### 4.1. Peroxisome Proliferator-Activated Receptors (PPARs)

Peroxisome proliferator-activated receptors (PPARs) represent a group of nuclear hormone receptors, which are expressed in various tissues influencing multiple systemic and metabolically relevant processes (β-oxidation of fatty acids, lipid transport, keto- or gluconeogenesis) [123]. PPARs can be classified in three isoforms: PPAR α, β/δ, γ with distinct but also complementary activities and functions potentially modulating NAFLD. PPARα is mainly found in fatty tissues, such as the liver. PPAR β/δ is widely distributed, with liver expression ranging from low to medium in humans and rats, and from medium to high in mice. On the contrary, PPAR γ is overexpressed in white adipose tissue and steatotic liver [124,125]. PPAR isoforms are not only distinctly expressed in different tissues, but also show differential profiles in different cell populations within tissues such as the liver (Figure 2).

#### 4.1.1. PPARα

PPARα directly regulates the cellular uptake of FAs, which is the classical process of FAs decomposition to produce energy [126,127]. Notably, PPARα can also regulate liver fat production, through the activation of sterol regulatory element binding protein 1c (SREBP1c) or indirectly coordinated via the LXR signaling pathway. PPARα-KO mice fed with high-fat diet (HFD) exhibit a large amount of hepatic lipid accumulation due to the inhibition of FAs uptake and oxidation. In addition, both high fructose fed mice and obese rats treated with selective PPARα agonists showed increased insulin sensitivity, indicating that PPARα is active at early stages of NAFLD [128]. Hepatic expression of PPARα correlates with presence, severity, and treatment response of NASH patients [129]. PPARα is predominantly involved in upregulation of hepatic gene expression levels related to gluconeo- and ketogenesis, mitochondrial and peroxisomal β-oxidation, as well as fatty acid binding and transportation [130]. Along those lines, experimental PPARα deletion in rodents aggravated hepatic fat storage [131]. On the contrary, it was demonstrated that adiponectin secretion and adiponectin receptor (AdipoR-1 and -2) expression was increased upon PPARα activation in adipocytes [132]. Wy-14,643–a PPARα agonist–increased AdipoR expression and decreased monocyte chemoattractant protein-1 (MCP-1) expression, ameliorating obesity related inflammation in adipose tissue as well as systemic insulin resistance in mice [133]. In addition to influencing steatosis, PPARα exerts anti-inflammatory properties, mainly by suppressing pro-inflammatory genes [134]. As a result, PPARα was shown to improve the pathology of NASH. In mice, MCD diet-induced steatohepatitis and fibrosis was reversed by applying the PPARα agonist WY-1 [127,135]. Activation of PPARα prevents lipid accumulation and inflammation in the liver by depressing the activation of pro-inflammatory macrophages, which ultimately leads to the reversal of typical NASH histological characteristics. Moreover, mice with deficiency of adipose triglyceride lipase produce few endogenous PPARα agonists. Compared with wild-type mice, they were more likely to develop steatohepatitis under the stimulation of LPS and MCD diet [136].

In addition to its anti-inflammatory properties, PPARα also plays an important role in liver protection by regulation of fibroblast growth factor 21 (FGF21). FGF21 can improve systemic insulin sensitivity and decelerate hepatic fibrosis. The activation of PPARα results in a significant increase in FGF21 in both liver and serum. Of note, PPARα- or FGF21- deficient mice fed the MCD diet were more prone to hepatic steatosis.

#### 4.1.2. PPAR β/δ

PPAR β/δ promotes fatty acid β-oxidation in extrahepatic tissues, is mainly expressed in adipose tissue acting as insulin sensitizer and prevents ectopic fat storage by induction of triglyceride formation of free fatty acids [137,138]. Furthermore, it regulates metabolic processes in multiple organs including the liver. In the liver, PPAR β/δ is expressed in hepatocytes, hepatic stellate cells (HSCs) and KCs, thus potentially participating in inflammation and fibrosis in NASH (Figure 2) [139]. A variety of monounsaturated fatty acids (MUFAs) actively bind to PPAR β/δ, regulating the metabolic homeostasis of FAs and glucose. PPAR β/δ mediates the synthesis of endogenous MUFAs via stearoyl-CoA desaturase 1 (SCD1). The activation of SCD1 eventually favors the liver protection via a positive feedback regulation [140,141]. In mouse liver, activation of PPAR β/δ can inhibit the expression of SREBP1c, which in turn was shown to attenuate liver steatosis [142]. Another mechanism by which PPAR β/δ inhibits hepatic steatosis and decelerates NAFLD progression was the regulation of low-density lipoprotein receptor (VLDLR). Multiple studies have shown that the expression of VLDLR is upregulated by PPAR agonists, including PPAR β/δ agonists. VLDLR levels and triglyceride accumulation is increased in PPAR β/δ knock-in mice, and vice versa decreased in PPAR β/δ knockout mice [143]. Conclusively, this indicates that PPAR β/δ is crucial for coordinating the VLDL-associated transcriptional response.

Besides regulating liver metabolism, PPAR β/δ also plays an important role in modulating inflammation. However, due to the inconsistent results from reports, the exact mechanism of PPAR β/δ in liver inflammation remains unclear. The activation of PPAR β/δ is related to the induction of anti-inflammatory signals and anti-inflammatory phenotype in KCs, which reduced metabolic disorders in the liver [144,145]. A more recent study comparing different PPAR agonists in experimental NASH in mice, revealed that PPARδ agonism and lanifibranor (pan-PPAR agonist) directly regulate the activation of liver macrophages, synergistically modulating NASH in addition to beneficial metabolic effects of PPARα/γ agonists [146]. PPAR β/δ-deficient mice treated with carbon tetrachloride (CCl_4_) showed more advanced liver fibrosis than wild-type mice. In CCl_4_-induced and bile congestion induced fibrosis models, the fibrogenesis of mice injected with PPAR β/δ agonists were improved [147,148]. Accordingly, PPAR β/δ is considered as a protective factor for liver fibrosis.

#### 4.1.3. PPAR γ

In the liver, PPAR γ promotes fatty acid oxidation and suppression of inflammatory pathways [149]. In addition, PPAR γ is also involved in extrahepatic effects regulating adipocyte differentiation, storage of fatty acids and glucose metabolism [150]. In mammals, PPAR γ consists of two isoforms (γ1 and γ2), both derived from one gene but different in length and expression. Generally, PPAR γ1 modulates cholesterol homeostasis, macrophage activation and suppression of inflammation. PPAR γ expression in the healthy liver is low, compared to adipose tissue, but in both patients and experimental animal models, elevated transcription levels of PPAR γ in the liver was a feature of steatosis [151,152]. Particularly, the hepatic expression of PPAR γ2 was shown to be induced by consumption of high caloric food [153]. Data indicate that PPAR γ induces liver lipid accumulation by promoting the synthesis of FAs and uptake. PPAR γ knockout prevented steatosis in HFD-fed mice and ob/ob mice [154,155]. PPAR γ binding to p65 attenuated the production of inflammatory cytokines and chemokines driven by NF-κB in the liver [156]. 

In macrophages, PPAR γ promoted an anti-inflammatory phenotype switch by up-regulating CD206 and CD163, which in turn inhibited the release of pro-inflammatory factors [157].

Fibrogenesis was alleviated by overexpression of PPAR γ in ob/ob mice via reducing HSCs proliferation, hepatic cycle arrest and apoptosis after MCD diet feeding for 2 months [158]. Activated HSCs may be reversed to a static phenotype, thus indicating that PPAR γ is able to regulate the expression of pro-inflammatory and profibrogenic genes [159,160]. The activity of HSCs exerted a beneficial impact in these studies, leading to the prevention of NASH. Moreover, blocking PPAR γ expression in macrophages and HSCs exacerbated the fibrotic response to CCl_4_-induced liver injury [149]. Ultimately, the absence of PPAR γ in macrophages exacerbated CCl_4_-induced liver fibrosis, as well as dietary induction of obesity and insulin resistance [146]. Thus it apprears that the effect of PPAR γ hepatocytes is steotogenic, promoting lipid droplets. However, the effect on other important cells in NASH including stellate cells and macrophages suggests a more regulatory role [161]. In aggregate, how PPAR γ drives or decreases liver damage is not entirely clear and might vary between species and individual patients.

### 4.2. Farnesoid X Receptor (FXR)

Farnesoid X receptor (FXR) is the most important NR for the maintenance of bile acid (BA) synthesis [162,163]. Subsequentially, fibroblast growth factor (FGF)-15/-19 axis is upregulated by FXR in enterocytes. FGF19 is an enterokine that reaches the liver via portal circulation and binds to the FGF receptor 4/β-klotho, which thereby represses bile acid synthesis and gluconeogenesis in liver [164]. Potentially, FXR-FGF-15/-19 regulation favors liver regeneration after injury [165]. In addition to bile acid metabolism, more and more has been discovered in recent years on the important role of FXR signaling in maintaining metabolic homeostasis of lipids, glucose and modulating immune responses [166,167,168,169,170,171]. Meanwhile, disruption of FXR signaling has been observed in many liver diseases, including NAFLD; FXR is thus considered a promising target for the treatment of NAFLD/NASH [172,173]. Intestinal FXR signaling and induction of FGF15/19 were shown to dampen steatosis, inflammation, and fibrosis as mice deficient in FGF15 fed a HFD had severe steatohepatitis [174,175].

FXR is expressed in many tissues, including kidneys, stomach, intestine, gall bladder, liver, and adipose tissue [176]. In obesity, modulation of FXR signaling proved to have beneficial effects [177]. Preclinical data regarding FXR in NASH are inconsistent. FXR-/- mice show elevated triglyceride and cholesterol levels in serum, along with excessive lipid enrichment in the liver [178]. In addition, FXR deficiency may also lead to insulin resistance [179,180]. In contrast, a recent study concluded that intestinal microbiota were important for promoting obesity in HFD fed mice in a FXR dependent fashion [181]. In this study, FXR-/- mice lacked the obese phenotype and liver damage, compared to wildtype mice fed HFD. In Zucker (fa/fa) rats, the FXR agonist obeticholic acid (OCA) activated FXR, eventually preventing steatosis, obesity, and insulin resistance [182]. The gut specific FXR agonist feraxamine showed improvement of obesity, insulin resistance and steatosis in obese mice [183]. In addition to metabolic modulation, NASH related histological features are comprehensively improved by FXR agonists, which reduce fibrosis and steatosis, as well as play anti-inflammatory roles. In MCD diet-fed mice, the FXR agonist WAY-362450 reduced liver inflammation and fibrogenesis without triglyceride enrichment [184,185]. In addition, a recent phase 2 clinical trial concluded that cilofexor, a small molecule FXR agonist, leads to reduction of steatosis and fibrosis in NASH patients [186]. In particular during later stages of NAFLD, bile acids levels increase and bile acid composition changes in patients, which is thought to be due to insulin resistance, intestinal dysbiosis and impaired hepatocyte excretion-together resulting in bile acid toxicity, further damaging the liver [13]. Hence, therapeutic modulation of FXR signaling might restore bile acid balance and thus explain some of the beneficial effects of FXR targeted therapeutics. 

### 4.3. Liver X Receptor (LXR)

Liver X receptor (LXR) has two isoforms, LXRα and LXRβ, and regulates hepatic triglyceride and cholesterol metabolism. In hepatic metabolism, LXR has a dual role: it regulates fatty acid metabolism by inducing expressions of SREBP1c, Stearyl-coenzyme A desaturase 1 (SCD1) and Fatty acid synthase (FASN). SREBP-1c is often considered to be a master regulator of lipid synthesis, along with which SCD1 and FASN also act as regulator in liver metabolism [119]. LXR signaling modifies energy storage by activating triglyceride synthesis and free fatty acid synthesis. However, LXR at the same time promotes cholesterol efflux, reduces cholesterol synthesis and uptake [161,187]. In NAFLD, LXR is thought to have opposing roles: while LXR expression was reported to increase with severity of NASH, potentially driving obesity and steatosis, LXR was also shown to suppress inflammation and improve hypercholesterolemia [188,189]. Therefore, the functions of LXR in NASH remain ambiguous. Small Ubiquitin-like modifier (SUMO) modified LXR suppresses the expression of several inflammatory genes (such as IL-1β and NOS) and the activation of NF-kB [190]. In a NAFLD mouse model, LXR activation prohibited the cascade of phosphoinosine-3-kinase (PI3K), reducing TNFα expression and liver damage [191,192]. In APO-E2 knock-in mice, LXR agonist treatment translated into depressed levels of cholesterol and inflammation, but elevated liver triglyceride levels [193,194]. In addition to the effects on metabolism, LXR has a role in modulating inflammation. In bone marrow-derived macrophages (BMDMs), the activation of LXR inhibited TLR2, TLR4, and TLR9 and related pathways, which resulted in anti-inflammatory features with less convergency of immune cells [188]. In NAFLD patients, LXR is upregulated in hepatocytes and monocytes [189]. Notably, in SREBP1c-defecient KCs, 27-hydroxycholesterol treatment reduced HFD-induced steatosis, leukocyte aggregation and pro-inflammatory gene expression [195]. In mice fed with a high-cholesterol diet, lack of LXRα promoted cholesterol accumulation, increased liver injury markers and KC activation [196]. In addition, in dendritic cells (DCs), LXR regulated cell migration through CCL19 and CCL21, thereby participated in leukocyte trafficking to lymph nodes [197]. Treatment with LXR agonists induced the differentiation of regulatory T cells (Treg) and inhibited the polarization of T helper cells (Th) 1 and Th17. All this evidence illustrates potentially protective effects of LXR in NASH. LXR deficiency of mice favors liver fibrogenesis and lipid accumulation [198,199]. Although several selective LXR agonists (such as Dessinosterol, GW6340 and LXRb agonist LXR-623) have been investigated, more data are required before therapeutical application in NAFLD [200].

### 4.4. Liver Receptor Homolog 1 (LRH-1)

Similar to other nuclear receptors, liver receptor homolog 1 (LRH-1) is a NR identified in the liver that exerts diverse metabolic functions including bile acid, lipid and glucose homeostasis [201]. Emerging evidence suggests a role for LRH-1 in NAFLD and NASH. Mice deficient in LRH-1 fed HFD had increased steatosis, liver injury and glucose intolerance, which was reversed by overexpression of LRH-1 [202]. Recently, the role of LRH-1 SUMOylation was implicated in the development of NAFLD [203]. Mice with a SUMOylation-defective mutant of LRH-1 (LRH-1 K289R) displayed early symptoms of NAFLD and NASH when challenged with different dietary models. The LRH-1 K289R mutation activated OSBPL3, thereby enhancing the activation of SREBP-1, resulting in de novo lipogenesis. In addition, in NAFLD mouse models and patients, the aggregation of OSBPL3-related inflammation and fibrogenesis indicates that the LRH-1-OSBPL3 signal may trigger the pathogenesis of NASH [201,203,204]. Furthermore, RNA-seq data from NAFLD or NASH patients showed that LRH-1 was significantly downregulated [205]. In summary, LRH-1 could be a promising target of NAFLD therapy.

## 5. Novel Strategies Targeting Metabolic Pathways in NAFLD

As awareness of NAFLD is increasing globally [206], ongoing preclinical and clinical studies provided various pathomechanistic insights into disease development and progression, identifying many new potential drug targets [24,207,208]. Lifestyle interventions, such as dietary changes or exercise, are still the major treatment recommendations for NAFLD, as no pharmacologic compound is approved yet. Although significant advances in drug development have been made over the last decade, data from clinical studies indicate limited efficacy of single drug treatments regarding patient-relevant endpoints (accepted endpoints in adults: resolution of NASH and/or improvement of fibrosis assessed on sequential liver biopsies) [209]. In this context, combinatorial therapeutic strategies targeting multiple pathways in parallel appear attractive and potentially bear significant advantages over single drug treatments (Figure 2) [210,211]. Targeting metabolic pathways, e.g., by targeting NRs, may not only improve steatosis and hepatic insulin resistance but might simultaneously influence the activation and polarization of various immune cell subsets (Figure 3). Early and promising trials have been published in recent years and will be outlined here (summarized in Table 1).

### 5.1. Peroxisome Proliferator-Activated Receptor (PPAR) Agonists

Members of the PPAR subfamily play an important role in mediating lipid metabolism in different tissues (Figure 3). Lanifibranor (IVA337) is a pan-PPAR agonist acting on all three PPAR isotypes, potentially combining pharmacological effects of each single PPAR, positioning lanifibranor as an interesting compound for basic research as well as for clinical trials. A translational study analysed therapeutic efficacy of lanifibranor in two independent mouse models of NASH and liver fibrosis, highlighting differential pharmacologic effects on metabolically altered hepatocytes, hepatic stellate cells (HSCs) and macrophages. Monocyte-derived macrophages (MoMF) were metabolically reprogrammed in NAFLD, displaying a distinct pro-inflammatory polarization state that was positively influenced by lanifibranor. Moreover, pan-PPAR agonism increased gene expression of *plin2* and *cd36* in MoMFs which facilitate lipid droplet formation reducing excessive, intrahepatic free fatty acids (FFA) content. In line, therapeutic administration improved lipid metabolism and subsequently hepatic inflammation, ameliorating steatohepatitis and fibrosis [146]. Lanifibranor was evaluated in a phase 2b clinical trial in adult patients with biopsy confirmed diagnosis of active NASH (NATIVE, ClinicalTrials.gov (accessed on 28 December 2021) Identifier NCT03008070). Results have recently been published reporting a significant higher proportion of patients with a decrease of at least 2 points in the activity part of the Steatosis, Activity, Fibrosis scoring (SAF-A) treated with lanifibranor at 1200 mg once daily over a period of 24 weeks, supporting subsequent evaluation in a phase 3 clinical trial [212]. Lanifibranor was also effective on several histological regulatory endpoints such as NASH resolution without worsening of fibrosis and the combining endpoint of NASH resolution and fibrosis improvement in the same patient.

Efficacy of elafibranor, a combined PPAR α/δ agonist, was tested in a phase 2 clinical trial in biopsy proven NASH patients (ClinicalTrials.gov (accessed on 28 December 2021) Identifier NCT01694849) (Figure 3). Of note, elafibranor did not meet primary endpoints (NASH resolution without fibrosis worsening) in the intention-to-treat analysis. However, post-hoc analysis revealed dose dependent beneficial effects in patients with a nonalcoholic fatty liver disease activity score (NAS) ≥4 modifying the primary endpoint of NASH resolution as disappearance of ballooning and reduced levels of lobular inflammation (score = 0 or 1) [213]. In addition, the authors reported favourable secondary effects: reduction of liver enzymes, improved cardiometabolic risk profile with decreased low-density lipoprotein-C (LDL-C) and elevated high-density lipoprotein (HDL) levels, improved glucose metabolism (fasting serum glucose, HbA1c, fasting insulin, HOMA-IR, and circulating free fatty acids) and reduced markers of systemic inflammation [213]. However, an interim analysis of a large phase 3 clinical trial did not meet the primary endpoint, resulting in the discontinuation of evaluating elafibranor in NASH.

Saroglitazar is another molecule combining pharmacological effects of PPAR α/γ agonism. Various translational studies demonstrated predominant effects on the PPARα. In experimental NASH mouse models, saroglitazar improved histological NASH features, serum aminotransferase levels as well as markers of inflammation and also prevented fibrosis. Of note, saroglitazar demonstrated additive beneficial effects compared to the single PPAR agonists (PPARα–fenofibrate and PPARγ–pioglitazone) [214]. Further preclinical data confirmed those findings, also indicating beneficial effects on obesity, dyslipidaemia and insulin resistance in independent mouse models [215,216]. In a phase 2 clinical trial (EVIDENCES IV, ClinicalTrials.gov (accessed on 28 December 2021) Identifier: NCT03061721), saroglitazar was tested in 106 patients with NAFLD/NASH for 16 weeks [217]. In this trial, saroglitazar significantly improved ALT, liver fat content and insulin resistance in patients [217]. A phase 2a, single center study is currently evaluating saroglitazar treatment in liver transplant recipients with NAFLD (EVIDENCES VIII, ClinicalTrials.gov (accessed on 28 December 2021) Identifier: NCT03639623).

MSDC-0602K is a novel thiazolidinediones (TZD) molecule preferentially acting as insulin sensitizer through binding to PPAR γ (Figure 3). TZD have been extensively investigated as antidiabetic agents and demonstrated various beneficial metabolic effects decreasing cardiovascular risk factors. Clinical use is still limited due to safety issues and negative side effects (edema, heart failure, elevation of liver enzymes, weight gain or hypoglycemia). Nevertheless, translational studies combine second-generation TZD such as MSDC-0602K with other compounds to further reduce the risk of negative side effects. Combination of MSDC-0602K and Omega-3 fatty acids showed additive effects on triacylglycerol and fatty acid cycling in adipose tissue [218] in an obesity mouse model while combination with the glucagon-like peptide-1 receptor agonist liraglutide improved glucose tolerance and histologic features of NASH better than the single drug application in experimental steatohepatitis [219]. EMMINENCE is a phase 2b clinical study that investigated treatment with MSDC-0602K in adult subjects with biopsy proven NASH and fibrosis not meeting the primary endpoint of histological improvement (ClinicalTrials.gov (accessed on 28 December 2021) Identifier: NCT02784444) [220].

### 5.2. Fibroblast Growth Factor (FGF) Analogues

Fibroblast growth factors are a family of 23 cell signalling molecules which are involved in various systemic signalling cascades comprising cell growth and differentiation as well as angiogenesis, wound healing processes and energy homeostasis. FGFs gain organ specificity through tissue distribution of the FGF receptors.

FGF19 is an intestinal expressed molecule involved in various metabolic pathways, including suppression of bile acid synthesis, enterohepatic bile acid circulation and cholesterol demand [221]. In a genetically altered diabetes mouse model, FGF15 (human ortholog, FGF19) administration stimulated hepatic glycogen synthesis improving glucose metabolism and insulin sensitivity [222]. Aldafermin or NGM282 is a humanized FGF19 analogue which significantly ameliorated histological features, imaging scores and serum markers in a phase 2b clinical trial in biopsy proven NASH patients (ClinicalTrials.gov (accessed on 28 December 2021) Identifier NCT02443116) [223]. However, aldafermin did not improve fibrosis stage in a biopsy-controlled phase 2b clinical trial in non-cirrhotic NASH patients, leading to the discontinuation of drug development in this indication [224].

FGF21 is a systemically circulating endocrine hormone expressed in various tissues. FGF21 agonists are mainly active in hepatic and adipose tissue due to FGF receptor expression profiles and tissue distribution of the co-receptor β-Klotho [225,226]. FGF21 stimulates glucose uptake in adipocytes thereby beneficially influencing insulin sensitivity and reducing hepatic fat content [227]. Therapeutic administration of an FGF21 analogue in a rodent dietary obesity model corrected bodyweight and diabetes [228]. A phase 2b clinical study in patients with NASH and stage 3 liver fibrosis treatment with pegbelfermin (BMS-986036)–a pegylated human FGF21 analogue–significantly reduced extent of hepatic steatosis and improved surrogate markers of NASH (FALCON1, ClinicalTrials.gov (accessed on 28 December 2021) Identifier NCT03486899) [229]. Efficacy and safety of pegbelfermin is currently also under evaluation in NASH induced cirrhotic patients (FALCON2, ClinicalTrials.gov (accessed on 28 December 2021) Identifier NCT03486912) [230].

### 5.3. Farnesoid X Receptor (FXR) Agonist-Obeticholic Acid (OCA)

Obeticholic acid (OCA) is a pharmacologic agonist binding the bile acid receptor FXR, that is mainly expressed in liver and kidney. OCA has been extensively investigated in NAFLD, and multiple beneficial metabolic effects were observed upon activation of FXR leading to reduction of hepatic bile acid concentrations (Figure 3). In line, FXR contributes to systemic metabolic processes regulating hepatic glycogenolysis, gluconeogenesis, de novo lipogenesis, fatty acid oxidation, as well as extrahepatic processes, such as insulin sensitivity in muscle and adipose tissue [221]. Administration of OCA for 72 weeks in adult NASH patients in a multicentre, randomised, placebo-controlled phase 2 trial effectively improved histological disease features (FLINT, ClinicalTrials.gov (accessed on 28 December 2021) Identifier NCT01265498) [231]. Post-hoc analysis demonstrated additional effects of OCA reducing obesity and serum aminotransferase levels. However, the authors observed increasing levels of alkaline phosphatase (ALP), low-density lipoprotein-C (LDL-C) and hemoglobin A1c levels (HbA1c) compared to placebo groups [232]. Subsequently, in the CONTROL study combination treatment with OCA and atorvastatin, which was safe and well tolerated, could reverse concerning elevation of LDL-C effectively (ClinicalTrials.gov (accessed on 28 December 2021) Identifier NCT02633956) [233]. In addition, pruritus is a common side effect of OCA. A phase 3 clinical trial with OCA in adult NASH patients is still ongoing and recently published interim analysis confirmed improvement of fibrosis while the primary outcome of NASH resolution was not met (REGENERATE, ClinicalTrials.gov (accessed on 28 December 2021) Identifier NCT02548351) [234].

### 5.4. Stearoyl-CoA Desaturase (SCD1) Inhibitor–ARAMCHOL

Aramchol is a synthetic bile acid (cholic acid) and fatty acid conjugate (arachidic acid) inhibiting the activity stearoyl-CoA desaturase 1 (SCD1)–an enzyme that catalyses the synthesis of monounsaturated fatty acids in the liver. From a mechanistic point of view, SCD1 inhibition could decrease fatty acid formation consequently reducing hepatic fat storage and improving insulin resistance. Those beneficial effects were observed in several experimental models leading to further assessment in clinical studies [235,236]. Results from a phase 2b clinical study in biopsy proven NAFLD revealed a good safety and tolerability profile of aramchol as well as significant and dose dependent reduction of hepatic fat content (ClinicalTrials.gov number NCT01094158 (accessed on 28 December 2021)) [237]. Based on those results the ARREST study analysed efficacy of aramchol in a total of 247 patients with biopsy confirmed NASH (NAS ≥ 4), overweight or obesity and prediabetes or type 2 diabetes (ClinicalTrials.gov (accessed on 28 December 2021) Identifier NCT02279524) [238]. At the highest dose of 600mg (compared to 400mg and placebo) aramchol was associated with decreased liver triglycerides and improvement of liver enzymes. In line, NASH resolution without worsening of fibrosis was achieved in 16.7% (600mg) compared with 5% in the placebo group and vice versa fibrosis improvement (by ≥ 1 stage) without worsening NASH was achieved in 29.5% (600mg) compared with 17.5% [238]. Although the primary endpoint was not statistically met, the promising results led to a phase 3 clinical trial. This multicenter clinical study is currently evaluating the efficacy and safety of aramchol in NASH fibrosis confirmed by liver histology (F1-F3) (ARMOR, ClinicalTrials.gov (accessed on 28 December 2021) Identifier NCT04104321).

## 6. Conclusions

NAFLD is an important and rising chronic liver disease that affects the global population and represents a major indication for end-stage liver disease and liver transplantation. In the coming decade, the burden of NAFLD will grow further due to (i.) a rising prevalence of obesity, metabolic syndrome, and type 2 diabetes and (ii.) a current lack of approved pharmacotherapies preventing disease progression. In addition, NAFLD is the fastest growing risk factor for hepatocellular carcinoma, which can develop even in the absence of cirrhosis. NAFLD is complicated by its multifactorial pathogenesis, and basic research and clinical trials over the past decades have demonstrated that drugs with a single mode of action, e.g., targeting a certain type of immune cell, are insufficient to address such a heterogeneous disease. Hence, novel treatment strategies are being developed, combining different pharmacological interventions to achieve patient relevant clinical endpoints. Nuclear receptors allow the organism to maintain metabolic flexibility in various cell types and different tissues. The different NRs have multiple effects by which they modulate altered metabolism and inflammation in NASH and thus, offer promising potential for therapeutic interventions. One caveat is that NRs are expressed by many tissues and were shown to have sometimes opposing functions, depending on the organ system and pharmacologic agent used. Therefore, more studies are required to address the simultaneous effects on metabolism and inflammation and in different tissues. In summary, as many new potential drug targets emerge, combinational treatment strategies might reveal additive beneficial effects in contrast to single drug treatments, helping to reach patient relevant endpoints and to sustain therapeutic benefits. In particular, combining metabolic, anti-inflammatory, and anti-fibrotic strategies represent a promising option to be addressed in future preclinical and clinical studies. Furthermore, stratifying patients at risk and individual disease stages could offer a more personalized treatment approach, which might improve clinical results.

## Figures and Tables

**Figure 1 ijms-23-02668-f001:**
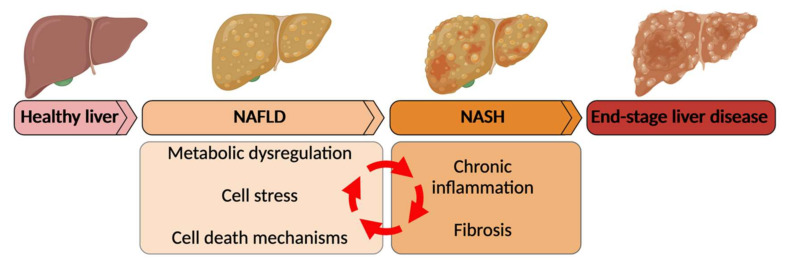
The spectrum of nonalcoholic fatty liver disease (NAFLD). NAFLD is a chronic liver disease that comprises a spectrum of metabolic liver diseases ranging from steatosis to nonalcoholic steatohepatitis (NASH) to end-stage liver disease including cirrhosis and hepatocellular carcinoma. Unique about the pathogenesis of NAFLD is the connected nature of the disease promoting events. Overnutrition leads to metabolic dysregulation resulting in cell stress and lipotoxicity. This triggers inflammation and recruitment of immune cells to the liver. Cell stress and inflammatory stress lead to hepatocyte death, which in NAFLD can be in different forms of cell death. (Created with Biorender.com, accessed on 23 February 2022).

**Figure 2 ijms-23-02668-f002:**
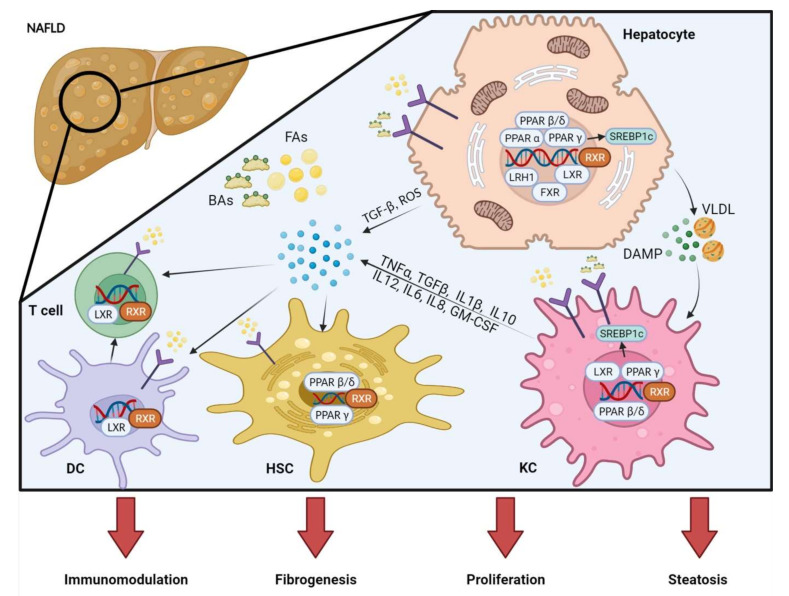
Nuclear receptors (NRs) modulate metabolism, inflammation, and fibrosis in NAFLD. Multiple NRs (e.g., PPAR α, PPAR β/δ, PPAR γ, FXR, LXR and LRH1) expressed by immune and liver parenchymal cells (hepatocytes, HSCs, macrophages, DCs and T cells) are activated by lipid deposition and bile acids. Binding RXR, NRs modulate lipid metabolism through downstream mediators (such as SREBP1c). In NAFLD, lipid-vesicle loaded hepatocytes secrete cytokines (such as TGFβ) and ROS to modulate the immune response. Hepatocyte-derived cytokines, FAs and Bas further shape the immune response and KC polarization. Coordinating with hepatocytes, activated KCs induce the activation of HSCs to myofibroblasts, which eventually promote fibrogenesis. In addition, activated DCs orchestrate the T cell immune response. NAFLD: non-alcoholic fatty liver disease; PPAR: peroxisome proliferator-activated receptor; FXR: farnesoid X receptor; LXR: liver X receptor; LRH1: liver receptor homolog 1; RXR: retinoid X receptor; SREBP1c: sterol regulatory element binding protein 1c; FAs: fatty acids; BAs: bile acids; HSC: hepatic stellate cell; KC: Kupffer cell; DC: dendritic cell; ROS: reactive oxygen species; TNF: tumor necrosis factor; TGF: transforming growth factor. (Created with Biorender.com, accessed on 23 February 2020).

**Figure 3 ijms-23-02668-f003:**
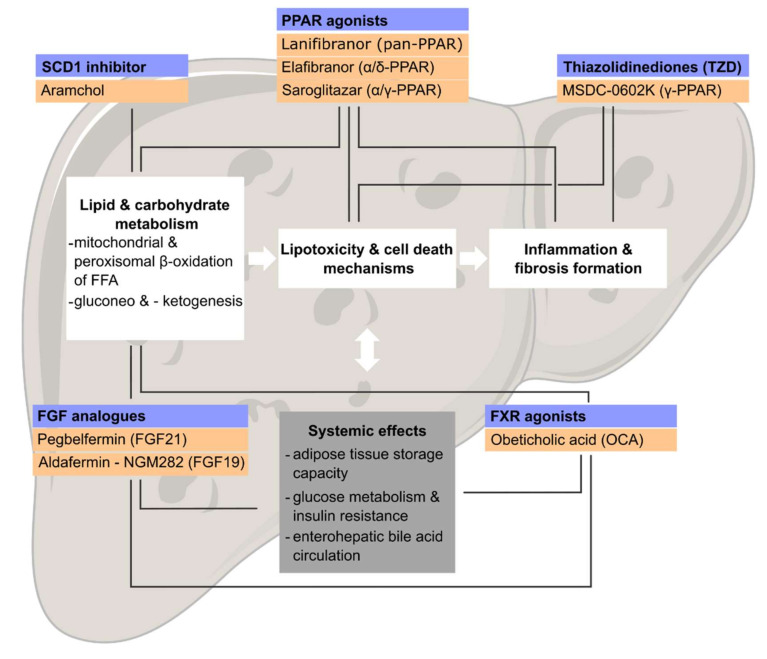
Novel pharmacologic compounds targeting metabolic and inflammatory pathways in NAFLD. During development and progression of NAFLD, multiple signalling pathways are dysregulated. Preclinical and clinical studies identified many potential drug targets for the treatment of nonalcoholic steatohepatitis and fibrosis. Targeting metabolic and inflammatory mechanisms of action (blue) represent promising strategies, which led to the development of various therapeutic compounds (orange), which are currently under clinical investigation. SCD1, stearoyl-CoA desaturase 1; PPAR, peroxisome proliferator-activated receptor; FGF, fibroblast growth factor; FXR, Farnesoid X receptor; FFA, free fatty acids.

**Table 1 ijms-23-02668-t001:** Selected pharmacologic compounds targeting nuclear receptors in Nonalcoholic fatty liver disease (NAFLD).

Pharmacologic Compound	Drug Target	Clinical Trials (ClinicalTrials.gov Identifier)	Phase
Lanifibranor	pan-PPAR agonist	NATIVE (NCT03008070)	phase 2b, completed
		NATiV3 (NCT04849728)	phase 3, recruiting
		NCT03459079	phase 2, recruiting
Elafibranor	PPAR α/δ agonist	NCT01694849	phase 2b, completed
		RESOLVE-IT NCT02704403	phase 3, terminated
Saroglitazar	PPAR α/γ agonist	EVIDENCES VIII (NCT03639623)	phase 2a, recruiting
		EVIDENCES IV (NCT03061721)	phase 2, completed
MSDC-0602K	PPAR γ agonist	EMMINENCE (NCT02784444)	phase 2b, completed
Aldafermin	FGF19 analogue	(NCT02443116)	phase 2, completed,
		ALPINE 2/3 (NCT03912532)	phase 2b, completed
Pegbelfermin	FGF21 analogue	FALCON1 (NCT03486899)	phase 2b, active not recruiting
		FALCON2 (NCT03486912)	phase 2b, active not recruiting
Obeticholic acid (OCA)	Farnesoid X receptor (FXR) agonist	FLINT (NCT01265498);	phase 2, completed
		CONTROL (NCT02633956);	phase 2, completed
		REGENERATE (NCT02548351)	phase 3, active, not recruiting
Aramchol	Stearoyl-CoA desaturase (SCD1) inhibitor	FLORA (NCT01094158);	phase 2, completed
		ARREST (NCT02279524);	phase 2b, completed
		ARMOR (NCT04104321)	phase 3, recruiting

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
