# Peer review of "Nuclear Receptors Linking Metabolism, Inflammation, and Fibrosis in Nonalcoholic Fatty Liver Disease"

_ijms, 2022, doi:10.3390/ijms23052668_

Round 1
Reviewer 1 Report
This is an extremely well written review highlighting role of nuclear receptors (NR) and NR-targeted drugs in NAFLD. The discussion of different mechanisms of cells death and immune cells is also important and timely. Some minor suggestions are provided to improve the review article. The authors also need to doublecheck references to make sure the right papers are cited. An example of mistaken referencing is provided below.
- Line 59 - change "those confounders" to "these factors".
- Line 110 - define the mouse model that is referred to here.
- Line 111 - replace the word devitalization with another more commonly used word.
- A table listing the clinical trials highlighted in the paper would be a nice addition.
- Carefully check all references for correctness. E.g., Line 650 - Refs. 168, 209 are not papers related to FGF15/19, but FXR.
Author Response
This is an extremely well written review highlighting role of nuclear receptors (NR) and NR-targeted drugs in NAFLD. The discussion of different mechanisms of cells death and immune cells is also important and timely. Some minor suggestions are provided to improve the review article. The authors also need to doublecheck references to make sure the right papers are cited. An example of mistaken referencing is provided below.
Response: We thank the reviewer for his/her positive evaluation of our manuscript and adjusted the manuscript accordingly.
- Line 59 - change "those confounders" to "these factors".
Response: We thank the reviewer for his/her comment adjusted the wording as suggested.
- Line 110 - define the mouse model that is referred to here.
Response: It was unfortunately not clear to us, which reference the author is referring to. In ll. 110 we outline the general principles of the pathogenesis of NASH in a few sentences, knowledge derived from many mouse models and human data.
- Line 111 - replace the word devitalization with another more commonly used word.
Response: We thank the reviewer for his comment and replaced “devitalization” with “exhaustion”.
- A table listing the clinical trials highlighted in the paper would be a nice addition.
Response: We would like to thank the reviewer his/her helpful suggestion. As recommended, we included “Table 1. Pharmacologic compounds targeting nuclear receptors in Nonalcoholic fatty liver disease (NAFLD)” in the manuscript demonstrating the different pharmacologic compounds and drug targets as well as the relevant clinical trials and stages.
- Carefully check all references for correctness. E.g., Line 650 - Refs. 168, 209 are not papers related to FGF15/19, but FXR.
Response: We thank the reviewer for his/her comment and great attention. We apologize for this mistake in our references and adjusted the references accordingly.
Reviewer 2 Report
Comprehensive and well conducted review.
Author Response
Comprehensive and well conducted review.
Response: We thank the reviewer for his/her very positive evaluation of our manuscript.
Reviewer 3 Report
ijms-1554890
This manuscript by Puengel and colleagues explores the role of nuclear receptors in NAFLD and NASH. Although quite interesting, the manuscript displays incoherence between the first and the last part. Indeed, while the first part analyse NAFLD and NASH from a more molecular point of view, depicting cellular death and inflammation, the second part in which nuclear receptors and their agonists are described do not take into account the mechanisms previously cited. Moreover, several inconsistencies are present in the section regarding NRs. All the concerns are listed below.
MAJOR ISSUES
- Manuscript should be more uniform, to underline better the implication of NRs in the major mechanisms behind NAFLD/NASH. In the present form, it is not clear how NRs could impact on cellular death and inflammation pathway that appear as crucial mechanisms of NAFLD/NASH.
- PPARg section. PPARg plays different role in the liver in healthy and diseased conditions. This should be displayed in the manuscript.
- FXR section. FGF15/19 should be mentioned.
- LXR section. Authors should focus on the dual role of LXRs: increasing FAs synthesis (via induction of SREBP1c, SCD1, FASN) while promoting cholesterol efflux. This will also explain better why the authors are discussing SCD1 inhibitor as a potential treatment for NAFLD.
- Line 646. FGF19 should not be discussed in the section regarding PPAR treatment.
- Line 695. Inhibition of SCD1 decrease only monounsaturated FAs formation, not all FAs.
MINOR ISSUES
- Line 27. Focusing.
- Important reviews on the topic have been recently published and should be included in the manuscript (PMID: 33545430; 33359400; 28442273)
Author Response
This manuscript by Puengel and colleagues explores the role of nuclear receptors in NAFLD and NASH. Although quite interesting, the manuscript displays incoherence between the first and the last part. Indeed, while the first part analyse NAFLD and NASH from a more molecular point of view, depicting cellular death and inflammation, the second part in which nuclear receptors and their agonists are described do not take into account the mechanisms previously cited. Moreover, several inconsistencies are present in the section regarding NRs. All the concerns are listed below.
Response: We thank the reviewer for his/her positive evaluation of our manuscript and the comprehensive comments and suggestions that definitely helped us to improve the manuscript.
Major issues:
- Manuscript should be more uniform, to underline better the implication of NRs in the major mechanisms behind NAFLD/NASH. In the present form, it is not clear how NRs could impact on cellular death and inflammation pathway that appear as crucial mechanisms of NAFLD/NASH.
Response: We absolutely agree with the reviewer here on his/her assessment of this issue. The key message of our manuscript is that there are many different avenues that promote NAFLD, including cell stress / cell death (section 2), inflammation and the effects of metabolism on inflammation (section 3). NRs are a neat family of receptors that play in all these areas and are thus promising targets in NASH. To provide better coherence, we have undertaken a few additional steps. 1. We have now added an additional Figure 1, a simple scheme that highlights the different pathomechanisms of NASH. 2. We have significantly rephrased section 4 (NRs) to allow the reader to logically follow why we designed the manuscript in that order. 3. We have added sentences to section 4 to fit better to the previous sections. 4. We discuss briefly NRs could impact cell death and inflammation.
- PPARg section. PPARg plays different role in the liver in healthy and diseased conditions. This should be displayed in the manuscript.
Response: We have slightly modified the PPARg section in the manuscript. Since the focus is really on NASH, we chose to highlight studies showing how PPARg is involved in steatosis in the liver.
- FXR section. FGF15/19 should be mentioned.
Response: We thank the reviewer for his/her expert opinion on this topic. We mentioned the fibroblast growth factor (FGF) -15 / -19 signalling pathway highlighting the close interconnection with the Farnesoid X receptor (FXR) signalling and regulation in the revised manuscript and also added some recent studies.
- LXR section. Authors should focus on the dual role of LXRs: increasing FAs synthesis (via induction of SREBP1c, SCD1, FASN) while promoting cholesterol efflux. This will also explain better why the authors are discussing SCD1 inhibitor as a potential treatment for NAFLD.
Response: We thank the reviewer for this critical comment. We have added two sentences in the section to address the comment.
- Line 646. FGF19 should not be discussed in the section regarding PPAR treatment.
Response: We thank the reviewer for his/her comment and adapted this section as suggested. We therefore excluded explanations regarding treatment with FGF19 analogues at this point separating the individual targeting strategies more clearly.
- Line 695. Inhibition of SCD1 decrease only monounsaturated FAs formation, not all FAs.
Response: We thank the reviewer for his/her expert comment and adjusted the manuscript accordingly clarifying that stearoyl-CoA desaturase 1 (SCD1) catalyses the synthesis of monoun-saturated fatty acids only.
Minor issues:
- Line 27. Focusing.
Response: This mistake was corrected.
- Important reviews on the topic have been recently published and should be included in the manuscript (PMID: 33545430; 33359400; 28442273)
Response: These articles are indeed timely and important and have been added to the list of references.
Round 2
Reviewer 3 Report
In my opinion, the new version of the manuscript is now ready for publication. I am satisfied with the changes made by authors.